# Clinical Characteristics of Neurocysticercosis in a Peruvian Population-Based Epilepsy Cohort: A Descriptive Cross-Sectional Study of Baseline Clinical Intake

**DOI:** 10.3390/pathogens12111313

**Published:** 2023-11-03

**Authors:** Samantha E. Allen, Luz M. Moyano, Melissa T. Wardle, Carolina Guzman, Sofia S. Sanchez-Boluarte, Gabrielle Bonnet, Javier A. Bustos, Seth O’Neal, Hector H. Garcia

**Affiliations:** 1Department of Neurology, University of California, San Francisco, San Francisco, CA 94143, USA; 2Center for Global Health Tumbes, Universidad Peruana Cayetano Heredia, Tumbes 24000, Peru; luzmariamoyano@gmail.com (L.M.M.); javier.bustos.p@upch.pe (J.A.B.); oneals@ohsu.edu (S.O.); hgarcia@jhu.edu (H.H.G.); 3Department of Epidemiology, Oregon Health and Science—Portland State University School of Public Health, Portland, OR 97236, USA; wardlem@ohsu.edu; 4Instituto Nacional de Ciencias Neurologicas, Lima 02002, Peru; carolina.guzman.d@upch.pe (C.G.); sofia.sanchez@upch.pe (S.S.S.-B.); 5Centre for the Mathematical Modeling of Infectious Diseases, London School of Hygiene and Tropical Medicine, London WC1H 9SH, UK; gabrielle.bonnet@lshtm.ac.uk; 6Department of Microbiology, School of Sciences, Universidad Peruana Cayetano Heredia, Lima 02002, Peru; 7Department of International Health, Bloomberg School of Public Health, Johns Hopkins University, Baltimore, MD 21224, USA; 8Department of Infectious Diseases, Oregon Health & Science University, Portland, OR 97236, USA

**Keywords:** epidemiology, epilepsy, neurocysticercosis

## Abstract

(1) Background: This study presents the baseline characteristics of a community-level population of people with epilepsy (n = 1975) living in an area endemic for *Taenia solium*, the pathogen responsible for neurocysticercosis (NCC). (2) Methods: Participants were sequentially enrolled in a clinical cohort from 2007 to 2020 in Tumbes, Peru. All participants provided demographic and clinical history and received clinical evaluations. Diagnostics, including neuroimaging, cysticercosis serologies, and EEG, were obtained where possible. The data presented are from the cross-sectional baseline assessment of cohort participants. (3) Results: Approximately 38% of participants met the criteria for NCC. Those with NCC were more likely to have adult-onset epilepsy, as well as a longer duration of epilepsy, as compared to their counterparts without NCC. Overall, the data indicate a large treatment gap, with only approximately a quarter of the baseline population with prescriptions for anti-seizure medications. (4) Conclusions: These data reveal a high proportion of NCC among people living with epilepsy in these communities, with limited health care resources. At baseline, 74% of the population were not receiving anti-seizure treatments. Further analyses of these data will clarify the natural history of the disease for this population.

## 1. Introduction

Neurocysticercosis is an infection caused by *Taenia solium*, commonly known as the pork tape worm, a parasite endemic to large swaths of Asia, Africa, and Latin America [1,2]. This infection is acquired through fecal-contaminated food or water. Once acquired, the larval cysts or cysticerci of *T. solium* can form throughout the body. The term neurocysticercosis (NCC) denotes cases in which the cysticerci form in the central nervous system. Development of cysticerci within the brain can lead to a wide range of neurologic morbidities, including seizure, intracranial hypertension, chronic headache, stroke, and cognitive impairment, as well as mortality [3].

The most common and notorious sequela of NCC is epilepsy [4]. Epilepsy attributable to both NCC and other etiologies is well known to cause significant disability and an elevated risk of mortality [4,5]. NCC is a frequent cause of acquired epilepsy in low-to-middle-income nations endemic for *T. solium* and is furthermore thought to underlie the finding that the incidence of seizures in Latin America is approximately twice as high as in high-income countries [6,7,8]. Despite this increased burden of disease, local communities have limited resources with which to address these infections and their associated neurologic morbidities as anti-seizure drug choices are limited and epilepsy surgery, a relatively common intervention for refractory epilepsy in North America and Europe, is not routinely available [9]. Overall, the prognosis of epilepsy among individuals with NCC at the time of their epilepsy diagnosis is still uncertain in the community setting, and further investigation is needed in order to inform effective epilepsy management for these individuals.

These gaps in knowledge have motivated the formation of a population-based epilepsy cohort in northern Peru to better understand this population and the disease itself, as well as to develop interventions aimed at addressing these infections. This study uses digitized medical records from a population-based epilepsy cohort assembled in northern Peru by the Cysticercosis Working Group in Peru. The goals of this study are to describe the baseline clinical characteristics of people living with epilepsy in northern Peru, describe the prevalence of NCC within this population, and assess differences in the distribution of participant characteristics based on baseline evidence of NCC. Through this exploratory analysis, we hope to draw out future lines of investigation, with the ultimate goal of understanding the clinical manifestations, treatment, and management of epilepsy in a setting highly endemic with *T. solium*.

## 2. Materials and Methods

Study Site: The study participants were primarily drawn from 107 villages within the state of Tumbes (population 81,170) [10]. Additional participants were identified from the urban centers of Tumbes, the neighboring state of Piura, and other surrounding areas as resources permitted. All areas included are known to be *T. solium* endemic. The cycle of transmission occurs when a human host becomes infected with the intestinal tapeworm through consumption of free roaming pigs who themselves consume *T. solium* eggs shed into the environment by another human tapeworm carrier. Many households lack latrines, and open defecation is common, making these infected human feces freely available to pigs. This fecal–oral cycle of infection in humans in what then results in cysticercosis and neurocysticercosis.

Study design and participants: Participants were screened through home visits for symptoms related to epilepsy, starting in 2006, using a validated questionnaire [11]. These screenings were focused within the 107 villages in the state of Tumbes noted above. Those who screened positive were referred from their local clinics. In addition to those captured through home visits, other participants self-referred to the clinic through word-of-mouth [4]. As needed, participants attended a clinical visit or received a home visit to receive care prior to their full clinical intake. Participants were enrolled in this prospective clinical cohort from 2007 to 2020, operating through the clinical unit of the Center for Global Health in Tumbes, Peru. The clinic was established to offer regionally accessible epilepsy-specific services including video EEG monitoring, on-site neuroimaging, and neurologic specialty consultations, as well as anti-seizure medications (ASMs) and ongoing epilepsy management.

Baseline participant information from 2007 to 2020 are included in the present study. This study specifically focuses on information collected during the initial intake appointment, rather than longitudinal data obtained over the course of participant follow-up. While the study population includes participants described in previous cross-sectional studies conducted in 2006 and 2007 [4] and has been used to inform modeling for neurocysticercosis [12], this is the first publication to describe the baseline characteristics of the full clinical cohort.

While detailed longitudinal analysis of the clinical cohort is out of scope of the present study, the extent of subsequent follow-up for participants is summarized in Appendix A and Appendix A. Over 90% of participants engaged in either a clinical follow-up or home visit and were followed for an average of 3 years (range 0–13 years). On average, there are approximately 8 clinical follow-up visits for each participant and 19 home visits.

Clinical Evaluation: Inclusion criteria included the follows: (1) greater than 2 years of age; (2) a diagnosis of epilepsy, defined as two or more unprovoked seizures separated by over 24 h; and (3) residence in a *T. solium* endemic region. Participants wishing to enroll consented to be followed as part of a clinical cohort according to the study protocol approved by the Institutional Review Board of the Universidad Peruana Cayetano Heredia, Lima, Peru. Membership in the cohort did not impact clinical care provided by the clinic. At the time of enrollment, all recruited participants completed a clinical questionnaire, including basic demographic information. Demographic variables collected included age, sex, and level of education. Basic medical history and parameters obtained included vital signs, height, and weight, as well as history of tobacco, alcohol, and substance use, and history of medical comorbidities. Body mass index (BMI) categories (underweight, normal, overweight/obese) were defined using the BMI formula (weight/height^2^) for individuals aged ≥20 years, and z-scores based on the World Health Organization (WHO) were used as reference standards for those <20 years. Lifestyle factors and medical comorbidities were noted as being present (yes/no) when participants self-reported a certain diagnosis. These included substance use (tobacco, alcohol, drugs) and medical comorbidities (diabetes, hyper-tension, dyslipidemia, heart disease, psychiatric diagnosis).

In addition to demographic information, participants were evaluated by the clinical unit of the Center for Global Health physician (LMM) to make/confirm a diagnosis of epilepsy and to collect further medical and epilepsy history. Basic information regarding epilepsy history collected during this visit included epilepsy risk factors, date of first lifetime seizure, date of last seizure, seizure semiology, frequency of seizures, diagnostics previously obtained, and treatments pursued. Epilepsy risk factors included a reported history (yes/no) of head trauma, a family member with epilepsy, neurosurgery, or developmental abnormality. Developmental abnormality included a reported history regarding any deviations from typical social, motor, language, and cognitive development [13]. Cases were independently reviewed by specialty trained neurologists to confirm a diagnosis of epilepsy and classify seizure and epilepsy type.

Once clinical history was collected, participants were asked to provide blood samples. An enzyme-linked immunoelectrotransfer blot (EITB or Western blot) using lentil lectin affinity-purified *T. solium* metacestode glycoprotein antigen was completed on all collected blood samples to aid in the diagnosis of NCC [14]. Antigen enzyme-linked immunosorbent assays (ELISA) were also run on a subset of the population [15]. The decision of whether or not to undertaking this testing was determined by the provider based on clinical diagnostic uncertainty. All participants were also offered a non-contrasted CT scan of the brain at the Center for Global Health (Tumbes, Peru). CT scans were interpreted by specialty trained radiologists. Participants requiring further diagnostics were referred for brain MRI to determine further treatment options (data not presented).

Routine sleep-deprived video EEG monitoring (20–30 min in duration), with provocative maneuvers including photic stimulation and hyperventilation, was also offered to all participants enrolled after 2015. The EEG testing was performed using a 16-lead EEG machine with electrodes placed according to the international 10/20 system. EEG results were interpreted by specialty trained neurologists.

Based on the described seizure semiology, with support from neuroimaging and EEG if available, participants’ seizures were classified as either focal onset, generalized onset, or unknown onset. Characteristics of focal-onset seizures included seizure aura, laterality of posturing/motor features, retained awareness, and/or evidence of progression to bilateral tonic–clonic activity. Generalized-onset seizures were assigned when the clinical history, neuroimaging, and/or EEG data were suggestive of generalized onset with semiologies consistent with tonic–clonic, clonic, tonic, myoclonic, atonic, or absence seizure activity.

A diagnosis of NCC was made using the principles of the Del Brutto criteria. Specifically, the results of neuroimaging, together with the clinical characteristics of the population, were used to confer a diagnosis of NCC [16,17]. All participants classified as having NCC met at least the criteria for “probable diagnosis” of NCC, and many participants within this group also met the stricter “definitive diagnosis” criteria as well. We chose not to distinguish among probable and definitive diagnostic groups in the analysis as limited clinical resources (e.g., limitation of MRI technology, difficulty obtaining serial imaging) made it difficult to obtain sufficient diagnostic information to allow for a definitive diagnosis to be made in many cases. Because of the reliance on imaging to make a diagnosis, participants without neuroimaging were excluded. All study participants meet the minor clinical/exposure Del Brutto criteria as they all have clinical manifestations—all have a diagnosis of epilepsy—and are residents of a cysticercosis endemic region. Participants with clear imaging findings consistent with NCC were given a diagnosis of NCC.

Imaging findings diagnostic of NCC included visualization of the presumed scolex (evidence of a hyperdense nodule within the interior of a parenchymal vesicular lesion), cystic lesions, enhancing lesion(s), multilobulated subarachnoid cystic lesions, and typical parenchymal brain calcifications. Calcified lesions were only classified as NCC-related if they had a typical appearance, pattern, and location, with typical appearance being small, clearly demarcated hyperdense rounded nodules or punctate lesions without perilesional edema [18]. Participants with calcifications in physiologic locations such as the falx or choroid plexus were excluded from the NCC group, as were participants with calcification patterns and locations consistent with other known disease states such as Fahr disease, arterial atherosclerosis, or a vascular malformation.

Participants who did not meet criteria for NCC as described above included individuals with both normal, non-lesional scans, as well as non-NCC related pathologies, including neoplasm, vascular lesions, ischemic changes, encephalomalacia, and lesions of uncertain etiology. Further characterization and classification of this non-NCC group is beyond the scope of this study.

All participants were offered epilepsy management in parallel with their participation in the research cohort through the Center for Global Health in Tumbes under the care of a fully licensed medical doctor with a special focus on epilepsy. This provider used the diagnostic work up obtained as part of the study to offer participants antiseizure treatment based on the current standard of care. Medications available through the Center for Global Health included carbamazepine, phenytoin, phenobarbital, valproate, clonazepam, and diazepam. Less commonly, participants would also be offered newer antiseizure medications including lamotrigine, topiramate, levetiracetam, gabapentin, and clobazam, although availability and cost were, in most cases, prohibitive.

Statistical methods: All data were analyzed using STATA SE17 (StataCorp, College Station, TX, USA). The main measure of occurrence was the prevalence of having NCC at baseline intake, which was estimated with a 95% confidence interval (CI). Descriptive statistics were used to characterize the participants overall and to compare the demographic, clinical, and epilepsy-related characteristics of participants with NCC to their non-infected counterparts. For categorical variables, differences in proportions were analyzed using a chi-square test, with a *p*-value of <0.05 considered statistically significant. For variables with smaller cell sizes, Fischer’s exact test was used. For continuous variables, a *t*-test was used. For non-parametric distributions, a Wilcoxon rank sum test was used. Insufficient answers were left as missing data in the database and were excluded from the analysis, and as a result, the number of individuals with available data for a specific variable might differ from the total number of study participants.

## 3. Results

A total of 1975 participants were enrolled in the epilepsy cohort. Of the participants, 90% (n = 1581) are from the state of Tumbes, with the remainder being from the state of Piura, and less than 1% are from Ecuador. Among participants from Tumbes, 882 were screened in their villages and subsequently diagnosed with epilepsy and enrolled in our clinical cohort. Of these 882 individuals, 689 are from rural areas and 193 are from peri-urban areas. The average age of the cohort was 29 years (range of 2 to 89 years). The population was nearly evenly split by sex, and most attained either primary (41%) or secondary education (39%) at baseline enrollment.

Regarding recorded health characteristics, less than 5% of participants reported tobacco use, about 20% reported regular alcohol consumption, and 0.3% reported drug use. The vast majority, almost 95% of the population, did not carry common medical diagnoses such as diabetes, hypertension, dyslipidemia, heart disease, or a psychiatric diagnosis (Table 1), although this is a population with limited access to the primary health care providers who typically make these diagnoses.

Most participants had active epilepsy at the time of baseline intake (86%), with focal epilepsy being the most common seizure type identified by history (55%). EEG was obtained in a little over 10% of the participants (n = 252). Among those with an EEG testing, approximately 41% were normal, another 29% had non-epileptiform abnormalities of cortical dysfunction, and the remainder had epileptiform findings (30%).

Among the 1691 participants with active epilepsy at intake, approximately a quarter of the population (n = 442) reported a history of ASM treatment. Most of these participants were currently taking ASMs (n = 425), while the remaining reported past or irregular use (n = 17). Almost all (96%) patients with active epilepsy at intake were indicated and prescribed ASM treatment.

Of the 442 individuals reporting a history of ASM use, the most commonly prescribed was carbamazepine; this medication was prescribed to nearly half of the population on ASMs. The next most common ASMs were valproate, phenytoin, and phenobarbital, in that order. Rarely were participants receiving newer-generation ASMs such as lamotrigine, levetiracetam, topiramate, and oxcarbazepine (Figure 1A). Among the 1523 participants who were prescribed ASMs at intake, 58% were prescribed carbamazepine, 25% phenytoin, and the remainder were prescribed other ASMs (Figure 1B).

CT scans were obtained from over 90% of participants (n = 1792). Figure 2 shows the diagnostic pathway by which participants were classified as having NCC or not. The estimated prevalence of NCC among participants with available CT scans was 37.9% (95% CI: 35.6%, 40.2%). Of the 1113 participants scans without evidence of NCC, 864 (77.6%) were non-lesional, and the remaining 249 (22.4%) had pathologic findings, including neoplasm, vascular lesions, ischemic changes, encephalomalacia, congenital malformations, and lesions of uncertain etiology.

Of the 679 participants with imaging findings consistent with NCC, most had calcified lesions (n = 586), while a much smaller number had evidence of viable cystic lesions (n = 42) or a mixture of both cysts and calcifications (n = 51). EITB studies were performed on nearly 90% participants (n = 1745), and of those tested, 33% tested positive for cysticercosis antibodies (n = 650) (Table 2). Relationship between serologies and imaging findings among those with NCC is shown in Appendix A. Of the 679 participants with findings compatible with NCC on neuroimaging, 363 had both major neuroimaging criteria and positive serologies. Using the Del Brutto criteria [17], the 363 participants with imaging findings and positive serologies met the criteria for a definitive diagnosis of NCC, and the remaining 316 met the criteria for a probable diagnosis of NCC.

In comparing the subgroups with NCC versus those without, asymmetries were noted (Table 3). People with NCC were, on average, older compared to the non-NCC group. Regarding epilepsy phenotype, participants with NCC were more likely to have adult-onset epilepsy as well as a longer duration of the disease compared to their counterparts without NCC (*p*-value < 0.001). Further, while those with active epilepsy and NCC were slightly less likely to have a prior history of ASM use, they were more likely to be prescribed ASMs at baseline when compared to those without NCC.

Unstratified analysis reveals a relative increased burden of NCC among alcohol and tobacco users and among participants with a diagnosis of hypertension; however, these findings, with the exception of alcohol use, are null when stratified by age (Appendix A). Notably, most participants with a reported comorbidity were in the 40+ years age group.

## 4. Discussion

This study reports the baseline characteristics of a large, community-based cohort of individuals living with epilepsy in northern Peru, a region highly endemic for NCC. Over a third of the population met the criteria for NCC, which not only supports the claim that this infection is the most common cause of acquired epilepsy in cysticercosis-endemic regions but also raises the possibility that this is the single most common cause of epilepsy overall in these areas. While prior studies have reported similar numbers, this is the largest study to date looking at this population [19,20,21]. Our findings, furthermore, support a relatively higher prevalence of epilepsy in cysticercosis endemic regions when compared to non-endemic regions [22].

This relatively higher prevalence is surely magnified by the staggering gaps in medical care—most notably, limited access to diagnostics and treatments. The vast majority of individuals had no prior imaging at the time of enrollment. While study personnel obtained CT scan imaging on approximately 90% of participants, MRI scans, the standard of care for epilepsy management in high income nations, were more difficult to obtain as there was no MRI scanner in the state, and the travel burden for participants could often not be justified.

Most notable for this population is the low rate of anti-seizure medication usage at the time of enrollment. Similar to prior reports looking at low-to-middle-income nations [23,24], only approximately one quarter of the population was receiving anti-seizure medication prior to baseline evaluation, translating to a treatment gap of 74%, where the term “treatment gap” is defined as the proportion of individuals with active epilepsy who are not receiving treatment. This treatment gap stands in stark contrast to the ASM treatment gap of less than 10% in high-income nations such as the United States of America and European countries [25].

To further magnify the problem, the treatment gap does not take into account the limited anti-seizure medication options available (mainly carbamazepine, valproate, phenytoin, phenobarbital, and benzodiazepines), meaning that even the participants who are receiving treatment are likely facing a greater burden of side effects, teratogenicity, and medication interactions than their counterparts in high-income nations, where medications such as levetiracetam, lamotrigine, and lacosamide, with more benign side-effect profiles, relative safety in pregnancy, and limited drug–drug interactions, are among the most popular prescriptions for individuals with epilepsy [26]. Further, this reported gap does not consider under-treatment, which we suspect would be higher in low-resource settings, where drug costs can be prohibitive and availability of care is limited [27].

Our data, furthermore, suggest a treatment gap beyond epilepsy. Hypertension, for example, the leading cause of cardiovascular disease and premature death worldwide, is believed to have a global prevalence of approximately 30% [28]; meanwhile, our population had a self-reported prevalence of approximately 8%. While it is possible that there is a lower-than-expected prevalence of hypertension in this population, this estimate is likely higher than reported as participants lack access to the basic medical services and continuity of care that would allow for such a diagnosis to be made. Similar discordances are evident for other diagnoses surveyed in this population, including diabetes, hypercholesteremia, heart disease, and psychiatric diseases.

Focusing on participant characteristics, many of the health and epilepsy characteristics aligned with expectations (e.g., equivalent prevalence of epilepsy and NCC by sex and a higher proportion of focal (as compared to generalized) epilepsies, as seen elsewhere). The relative increased burden of NCC among alcohol and tobacco users and among participants with a diagnosis of hypertension became null in the post age-stratified analysis, with alcohol use being an exception, indicating that age is likely a confounder or effect measure modifier in the relationship between NCC and various comorbidities.

A more thorough characterization of this population will be made possible by years of follow-up data, as the data presented here look only at the baseline characteristics of participants at the time of enrollment. Through the follow-up data, we hope not only to better characterize the disease course and how it compares between participants with NCC and those without, but also to look at the determinants of medication adherence, mortality, the risk factors for refractory disease, the success of medication withdrawal trials, quality of life, and barriers to disease control, as well as time to disease control, for those with medically responsive seizures.

The strengths of this study include the population-based nature of recruitment in this region endemic for NCC, allowing for a better understanding of the true characterization of those with epilepsy and the breadth of the impact. Given a total population of 81,170 among the villages where enrollment was focused within the states of Tumbes [10], and using the estimated 17.25 per 1000 people with epilepsy found in prior community-based surveys [4], we would expect a total of approximately 1400 individuals with epilepsy to be living in this region. Given that we enrolled at least 882 participants from these rural and peri-urban areas, we estimate that we have captured 64% of the expected epilepsy population in this region. A further strength of this study is the large size of the cohort and the duration of follow-up, which will allow for a clear characterization of the disease course and participant outcomes.

The limitations for this study reflect many of the limitations faced by clinicians practicing in this setting: there is limited availability of diagnostics, including imaging capability, particularly MRI, as well as EEG, an important test used in the diagnosis of epilepsy. Without ready access to many of the typical modalities of testing used for the diagnosis of NCC as outlined in the Del Brutto criteria [17], such as plain film radiographs, serial imaging, tissue biopsy pathology, MRI, and ophthalmologic exams, it is possible that certain participant diagnoses have been miscategorized. Given that there is no single diagnostic modality that can identify all cases of NCC, we needed to be practical in classifying participant diagnoses. For example, we designated those with positive imaging findings but negative serologies as having NCC since prior studies have shown that less than a third of patients with a single lesion will have a positive EITB result, and furthermore, the sensitivity of serologic testing considerably decreases in patients with calcified lesions [29,30].

We further acknowledge the limitations of the diagnostics available to our research and clinical team. Despite our best efforts to classify participants based on CT findings, this imaging modality has its own limitations as cases could be missed, and there also exist mimics of NCC such as toxoplasmosis, echinococcus, and tuberculosis, which could result in misclassification of participants.

Another limitation is that it is possible that our cohort is enriched with participants with NCC beyond what would be expected on a population level as the Center for Global Health in Tumbes is known to care for patients with this condition and attracts these patients.

A further limitation is that clinicians used non-standard definitions for some of the participant-reported historical data, making the measurements of these particular variables more difficult to directly compare to other populations—most notably, family history of seizures and history of febrile seizures. Family history of seizures was reported as positive not just for those with first-degree relatives with a history of seizure but also if any relative, regardless of degree, was reported to have a history of seizure (Table 1 and Table 2). The febrile seizure category also included a reported history of prolonged febrile illness in childhood to capture individuals at risk for neurosepsis (Table 3). These definitions likely help account for the higher-than-expected reported prevalence of the epilepsy risk factors in this population. Despite this higher-than-expected prevalence, we acknowledge the limitations of brain imaging with CT technology and realize that we may have been unable to diagnose extra-parenchymal NCC, mild NCC, and certain cases of viable parenchymal cysts (which are difficult to see on CT [31]), as well as in cases of participants whose lesions had resolved.

Finally, while we have reported the prevalence of NCC within our population, we cannot know whether or not a participant’s epilepsy is truly attributable to NCC, given the high baseline infection rate in this setting. While serologies can be somewhat helpful in narrowing the pool, participants with 1–2 bands rather than 3+ bands may simply represent individuals with aborted infections rather than true infections [29]. The cross-sectional nature of these data does not allow us to describe the incidence of new onset epilepsy for the cohort participants; thus, our prevalence estimates must be interpreted with caution.

## 5. Conclusions

In summary, this study presents the baseline characteristics of a cohort of nearly 2000 participants with epilepsy living in a cysticercosis-endemic region with a high proportion of NCC (38%) and relatively low baseline access to health services. This study demonstrates the treatment gap in this population and further motivates efforts directed at prophylaxis as well as the dissemination of information in this region. In addition to highlighting this treatment gap, this study also allows for hypothesis generation around risk factors for NCC. Data collected from years of cohort follow-up will be forthcoming and will better clarify the disease course for these participants, including associated morbidity and treatment response over time.

## Figures and Tables

**Figure 1 pathogens-12-01313-f001:**
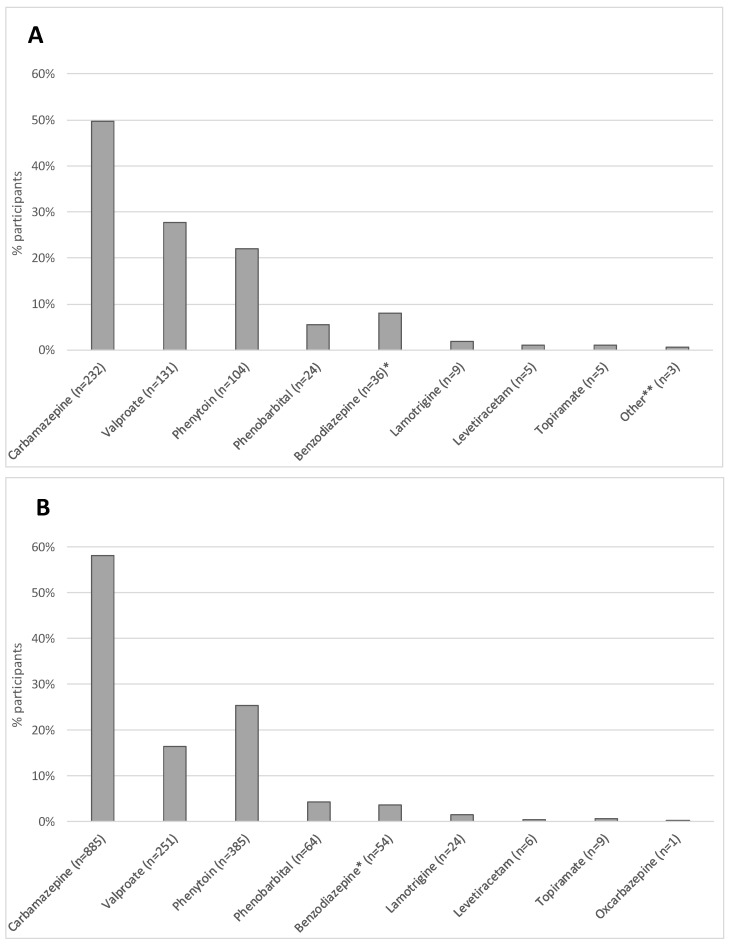
Anti-seizure medication (ASM) type among 442 participants who reported a history of ASM (panel (**A**)) and among 1523 participants who were prescribed ASM at baseline intake visit (panel (**B**)). Note: participants receiving more than one prescription may be duplicated. * Includes any benzodiazepine class of medications, including clobazam, clonazepam, and others; ** includes oxcarbazepine, gabapentin, bromide, and medication unknown.

**Figure 2 pathogens-12-01313-f002:**
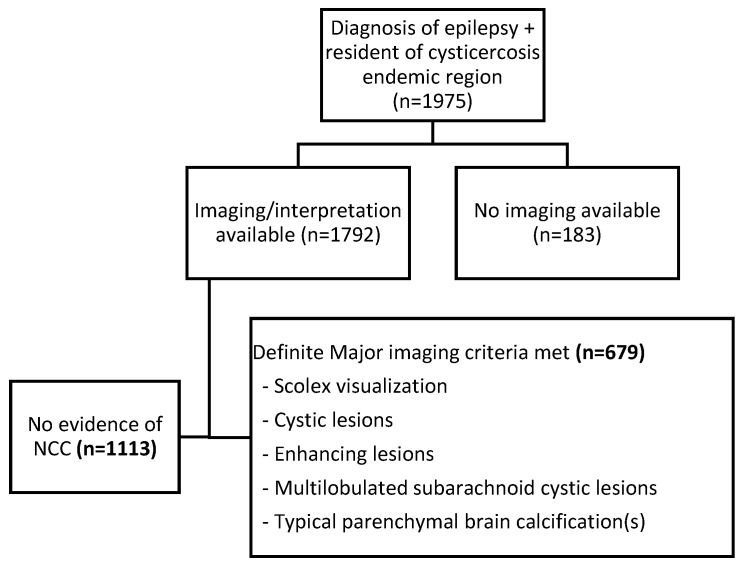
Flowchart showing diagnostic pathway for neurocysticercosis (NCC).

**Table 1 pathogens-12-01313-t001:** Baseline demographic, health, and epilepsy characteristics among participants enrolled in the Tumbes Epilepsy Clinical Cohort, 2007–2020 (N = 1975).

	N = 1975	
	n	%
Demographics		
Age, years		
Average (SD)	29.4 (16.9)	
Sex		
Male	966	48.9
Female	1009	51.1
Educational attainment		
No formal education	83	5.1
Primary	707	40.7
Secondary	634	39.2
Trade or other vocational	30	1.9
Tertiary	165	10.2
Health Characteristics		
BMI, kg/m^2^		
Underweight	50	2.5
Healthy	626	31.7
Overweight or obese	526	26.6
Missing	773	39.1
Current substance use		
Tobacco ^†^	78	4.0
Alcohol ^†^	408	20.7
Drugs ^†^	6	0.3
Previously diagnosed medical comorbidities		
Diabetes ^†^	23	1.6
Hypertension ^†^	56	2.8
Dyslipidemia ^†^	34	1.7
Heart Disease ^†^	14	0.7
Psychiatric diagnosis ^†^	38	1.9
Epilepsy risk factors		
Head Trauma ^†^	12	0.6
Febrile seizure or fever lasting > 15 days ^†^	725	36.7
Family history of epilepsy ^†^	1127	57.1
History of neurosurgery ^†^	15	0.8
Developmental abnormality ^†^	225	11.4
Epilepsy Characteristics		
Active epilepsy *		
Active	1691	85.6
Inactive	181	9.2
Missing	103	5.2
Age of onset		
<18 years	1073	54.4
18 or older	586	29.7
Missing	316	16.0
Duration of epilepsy, years		
Average (SD)	13.2 (13.4)	
Median (IQR)	9.0 (2–20)	
Missing	303	15.4
Seizure types by history		
Generalized	400	20.3
Focal	1084	54.9
Indeterminate	38	1.9
Missing	453	22.9
EEG results (n = 252) **		
Normal	102	40.5
Non-specific abnormality	73	29.0
Focal epileptiform	34	13.5
Generalized epileptiform	21	8.3
Epileptiform uncertain focal versus generalized	22	8.7
ASM Characteristics (n = 1691) ***		
History of any prior ASM usage reported		
Yes	442	26.1
Number of prior ASM(s) used ****		
Monotherapy	372	84.2
Dual therapy or more	70	15.8
ASM prescribed at baseline intake		
Yes	1523	96.0
Number of ASM(s) prescribed at baseline intake ****		
Monotherapy	1377	90.4
Dual therapy or more	146	9.6

Abbreviations: SD = standard deviation; BMI = body mass index; ASM = anti-seizure medication; EEG = electroencephalogram. * Seizures reported within the past year; ** EEG machine unavailable until 2015; *** among those with active epilepsy; **** among those using or prescribed ASM. ^†^ Among 1975 participants, data were missing for age (n = 2; 0.1%) and BMI (n = 773; 39.1%).

**Table 2 pathogens-12-01313-t002:** Cysticercosis and neurocysticercosis (NCC) diagnostic assessment.

	n	%
Diagnostics		
CT Scan Result (n = 1792)		
No NCC	1113	62.1
Calcified NCC	586	32.7
Viable/colloidal NCC cysts	42	2.3
Cysts and calcifications	51	2.9
Serologies		
EITB assay results (n = 1745)		
0 bands	1095	62.8
1 or more	650	37.3
Antigen ELISA results (n = 176)		
Negative	122	30.7
Positive	54	69.3

Abbreviations: CT = computed tomography; EITB = enzyme-linked immunoelectrotransfer blot techniques; ELISA = enzyme-linked immunoassay; NCC = neurocysticercosis.

**Table 3 pathogens-12-01313-t003:** Comparison of baseline characteristics among participants with and without NCC (n = 1792).

	NCC N = 679		Non-NCC N = 1113		
	n	%	n	%	*p* Value
Demographics					
Age, years					
Average with SD	34.3 (16.9)		27.0 (16.3)		<0.001
Sex					
Male	333	49.0	545	49.0	0.975
Female	346	51.0	568	51.0	
Health Characteristics					
BMI, kg/m^2^					
Underweight	11	1.6	32	2.9	0.027
Healthy	189	27.8	368	33.1	
Overweight or obese	188	27.7	281	25.3	
Substance Use					
Tobacco ^†^	36	5.3	35	3.1	0.02
Alcohol ^†^	163	24.0	201	18.1	0.002
Epilepsy Risk Factors					
Head Trauma ^†^	1	0.2	11	1.0	0.037 ^+^
Febrile seizure or fever lasting > 15 days ^†^	197	29.0	447	40.2	<0.001
FH of epilepsy ^†^	380	56.0	646	58.0	0.329
History of neurosurgery ^†^	3	0.4	8	0.7	0.549 ^+^
Developmental abnormality ^†^	51	7.5	136	12.2	0.002
Epilepsy Characteristics					
Active epilepsy (n = 1713) *					
Active	572	88.0	971	91.4	0.025
Inactive	78	12.0	92	8.7	
Age of Onset (n = 1541)					
<18 years	338	57.6	661	69.3	<0.001
18 or older	249	42.4	293	30.7	
Duration of Epilepsy (n = 1553)					
Average (SD)	15.8 (15.0)		12.1 (12.3)		<0.001
Median (IQR)	12.0 (3–25)		8.0 (2–19)		
Seizure types by history (n = 1412)					
Generalized	112	21.3	250	28.1	0.009
Focal	397	75.5	623	70.0	
Indeterminate	17	3.2	17	2.0	
EEG results (n = 238) **					
Normal	35	46.7	60	36.8	0.331^+^
Non-specific abnormality	20	26.7	49	30.1	
Focal epileptiform	12	16.0	21	12.9	
Generalized epileptiform	3	4.0	16	9.8	
Epileptiform uncertain focal versus generalized	5	6.7	7	10.4	
ASM Characteristics (n = 1543) ***					
History of any prior ASM usage reported					
Yes	131	22.9	266	27.4	0.051
History of number of prior ASM(s) used ^†^					
Monotherapy	111	84.7	221	83.1	0.676
Dual therapy or more	20	15.3	45	16.9	
ASM prescribed at baseline intake					
Yes	539	94.2	859	88.5	<0.001
Number of ASM(s) prescribed at baseline ^†^					
Monotherapy	503	93.3	763	88.8	0.005
Dual therapy or more	36	6.7	96	11.2	
Cysticercosis Diagnostic Serologies					
Serologies					
EITB assay results (n = 1619)					
0 bands	281	43.9	745	76.1	<0.001
1–2 bands	95	14.8	110	11.2	
3+ bands	264	41.3	124	12.7	
Antigen ELISA results (n = 157)					
Negative	59	59.0	49	86.0	<0.001
Positive	41	41.0	8	14.0	

Abbreviations: NCC = neurocysticercosis; SD = standard deviation; BMI = body mass index; ASM = anti-seizure medication; EEG = electroencephalogram; EITB = enzyme-linked immunoelectrotransfer blot techniques; ELISA = enzyme-linked immunoassay. * Seizures reported within the past year; ** EEG machine unavailable until 2015; *** among those with active epilepsy; **** among those using or prescribed ASM. ^†^ Conditional on those who reported ASM history or were prescribed ASMs. ^+^ Fisher’s exact test was used to account for cell size (<5).

## Data Availability

The data presented in this study are available on request from the corresponding author. The data are not publicly available due to participant privacy.

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
