# Peer review of "Clinical Characteristics of Neurocysticercosis in a Peruvian Population-Based Epilepsy Cohort: A Descriptive Cross-Sectional Study of Baseline Clinical Intake"

_pathogens, 2023, doi:10.3390/pathogens12111313_

Round 1

Reviewer 1 Report

Comments and Suggestions for Authors

In the introduction section, it is mentioned that “NCC is the number one cause of acquired epilepsy in low to middle income nations (LMIC)”. This statement is not accurate. I invite the authors to refer to the ILAE’s Prevention Task Force publication (Thurman DJ, et al. The primary prevention of epilepsy: A report of thePrevention Task Force of the International League Against Epilepsy. Epilepsia. 2018;59(5):905-914). This publication reports that stroke and infectious diseases are more frequent etiologies of epilepsy in LMIC. The paper also states that “Among some rural LMIC communities, the median proportion of epilepsy cases attributable to endemic NCC was 34%” However, this figure was calculated based on just three studies conducted in highly endemic areas. Furthermore, the authors advised caution in interpreting this high proportion, as “most of these studies examined prevalent cases of epilepsy where temporal relationships between NCC infection and seizure onset could not be determined”

A basic epidemiological concept for studying the etiology of a chronic disease is to select incident cases (i.e. new-onset epilepsy). When choosing, a sample of prevalent cases of epilepsy, it is not possible to determine which pathology appeared first, making it difficult to distinguish among the potential etiological factors that preceded the onset of epilepsy. Therefore, the question that arises in this study is which appeared first, epilepsy or NCC, especially in a highly endemic population for taeniasis/cysticercosis. The authors report that 85.6% of the selected population has “active epilepsy”, (defined as seizures reported within the past year), and 181 had inactive epilepsy, indicating that the majority of the selected population had chronic epilepsy. This methodology is the main drawback of this study and hinders the obtainment of valid results.

In the methodology section, concerning the diagnosis of NCC,  the authors did not to distinguish between probable and definitive diagnosis of NCC. This lack of distinction is misleading, as it appears that the majority of patients were diagnosed as probable NCC. If this is the case, the title of the manuscript might be more appropriately labeled: “Clinical characteristics of probable neurocysticercosis in a …….” 

Additionally, most of the patients diagnosed with as NCC had only calcifications (86% of the NCC sample). This is also debatable since calcifications  are not unique to NCC. Many other pathologies can result in similar calcifications, such as vascular, neoplastic, endocrine, viral, as well as other parasitic and infections diseases ( Grech R, et al. Intracranial calcifications. A pictorial review. Neuroradiol J 2012;25:427–51)

It´s also essential to inquire about CT scan results for the remaining 1133 patients who supposedly did not have NCC. The authors surprisingly omitted this important information, which is necessary for a comprehensive assessment of the background of epilepsy in the selected population, and for making comparisons with other similar populations.

Reviewer 2 Report

Comments and Suggestions for Authors

Line 169 “Findings included visualization of the scolex, cystic lesions, enhancing lesion(s), multilobulated subarachnoid cystic lesions, and typical parenchymal brain calcificationsIn what kind of materials and what method was used to prove the scolexes? Scolexes can be seen only with a microscope. What are the sizes of calcifications in cases accepted as NCC and are the calcifications in the brain the same size? If the lesions are of different sizes, it is probably cystic echinococcosis.

Recommendation: To add in the conclusion that prophylactic measures in endemic areas are needed and also dissemination of information about the cysticercosis disease.

Reviewer 3 Report

Comments and Suggestions for Authors

This article is very interesting, well thought out and organized.

In my opinion there are only minor points that can be improved.

Firstly regarding Del Brutto's criteria and his bibliography in general, there are two recent articles that deserve to be cited: 1) Del Brutto OH. Twenty-five years of evolution of standard diagnostic criteria for neurocysticercosis. How have they impacted diagnosis and patient outcomes? Expert Rev Neurother. 2020 Feb;20(2):147-155. doi: 10.1080/14737175.2020.1707667. Epub 2019 Dec 25. PMID: 1855080. 2) Del Brutto OH. Human Neurocysticercosis: An Overview. Pathogens. 2022 Oct 20;11(10):1212. doi: 10.3390/pathogens11101212. PMID: 36297269; PMCID: PMC9607454.

Another point that could be important to develop concerns neurocysticercosis in pregnancy. A brief mention of the effects on the pregnant mother and the fetus, how the cysticercus overcomes the blood-placental barrier. Since cysticercosis can induce intrauterine fetal death, possibly describe evidence of neurocysticercosis in intrauterine MRI

Reviewer 4 Report

Comments and Suggestions for Authors

Dear Authors,

I have read with great interest your manuscript entitled "Clinical Characteristics of Neurocysticercosis in a Peruvian Population Based Epilepsy Cohort: A Descriptive Cross-Sectional Study of Baseline Clinical Intake". Although it assessed an extremely important neurological issue caused by pathogens, which is underestimated as a potential risk factor for epilepsy, the manuscript needs further development before it can be published. I would suggest including more references in the Discussion section, where whole paragraphs are lacking references. The last paragraph of this section is purely hypothetical. Furthermore, a more clear definition of the aims of this study should be provided. You are reflecting on developing "interventions aimed to address these infections", but the manuscript does not address this issue. 

 I would recommend a more thorough statistical analysis of the data, before drawing any conclusion on how age affects NCC or associated comorbidities. You should briefly explain the decision to divide the population into three age categories (< 20 years;20-39 years; 40+ years). 

Another important concern observed by this reviewer is the lack of a proper definition of some comorbidities and epilepsy risk factors. For example, it would be interesting to know what kind of developmental abnormalities occurred in NCC and non-NCC patients. Furthermore, some psychiatric conditions may also be associated with epilepsy, but more detail should be provided to clarify the possible links between them.

Comments on the Quality of English Language

The manuscript also needs moderate language editing before publication.

Round 2

Reviewer 1 Report

Comments and Suggestions for Authors

Response 1: We thank you for helping to increase the precision of our word choice, and have accordingly changed the sentence to say “NCC is a major cause of acquired epilepsy in low to middle income nations” on page 2, line 56. We have also added the Thurman reference. 

Once again, this statement is not true. The IALE’s Prevention Task Force review (Thurman et al) concludes that stroke and infectious diseases are more frequent etiologies of epilepsy in most of LMIC, except in very few reports from highly endemic areas, in which “the median proportion of epilepsy cases attributable to endemic neurocysticercosis was 34%”.  I’d recommend the following phrase: “NCC is a frequent cause of seizure/epilepsy in endemic areas for T/S” Otherwise, the authors are misinforming the epidemiology of epilepsy.

Response 2: We agree that a better assessment of the etiologies of epilepsy can be obtained from incidence studies. However, this manuscript presents a large cohort of PWE in a rural setting, where only limited diagnostic capabilities exist, and on the view of the findings, assesses them in terms of the most frequent lesions on neuroimaging, images compatible with NCC. We explicitly state that we are not studying the etiology of epilepsy within this population. This study provides initial, exploratory evidence about a population of people with epilepsy who are typically excluded from studies, but we still recognize the limitation of our available diagnostics and mention this in our limitation section. 

The authors agree that a “better assessment of the etiologies of epilepsy can be obtained from incidence studies”; however, it is no even mentioned in the revised manuscript

The authors mention that the CT scans performed in 1113 (62%) patients with epilepsy did not have NCC.  Therefore, the obvious question is why the etiology of epilepsy in these patients could not be established? What were the CT scan findings in these patients?

Response 4: The participants scans were reviewed by expert radiologists with experience distinguishing calcifications likely attributable to NCC versus other etiologies. Calcified lesions were only classified as NCC related if they had a typical appearance, pattern, and location(s). Typical appearance being small, clearly demarcated hyperdense rounded nodules or punctate lesions without perilesional edema. Participants with calcifications in only physiologic locations such as the falx or choroid plexus were excluded from the NCC group as well as participants with calcification patterns and locations consistent with other known disease states such as Fahr Disease, arterial atherosclerosis, or a vascular malformation. This clarification is added to the methods section on page 4, line 209-223. We acknowledge that this method is imperfect in our limitations section (page 12, line 399) and some participants may be misclassified. 

The statement “Typical appearance being small, clearly demarcated hyperdense rounded nodules or punctate …..” is still debatable, since there are many others pathologies with similar lesions, such as toxoplasmosis, tuberculosis, tuberous sclerosis, etc. This another limitation to be mentioned in the manuscript

Reviewer 4 Report

Comments and Suggestions for Authors

Dear authors,

Thank you for your kind responses to the concerns raised by me previously. However, I think that your Response 2 should be included in the manuscript, letting readers understand your methodology. Moreover, I suggest putting the stratified analysis results in the Results section, not in the Discussion.
